**Data Availability Statement:** The data contains potential identification of each study participant,

# Patients' perceptions regarding multidrug-resistant tuberculosis and barriers to seeking care in a priority city in Brazil during COVID-19 pandemic: A qualitative study

Felipe Lima dos Santos[1]*, Ludmilla Leidianne Limirio Souza[1], Alexandre Tadashi Inomata Bruce[1], Juliane de Almeida Crispim[1], Luiz Henrique Arroyo[1], Antônio Carlos Vieira Ramos[1], Thaís Zamboni Berra[1], Yan Mathias Alves[1], Alessandro Rolim Scholze[1], Fernanda Bruzadelli Paulino da Costa[1], José Francisco Martoreli Júnior[1], Ana Carolina Scarpel Moncaio[2], Ione Carvalho Pinto[1], Ricardo Alexandre Arcêncio[1]

1 Department of Maternal-Infant and Public Health Nursing, University of São Paulo at Ribeirão Preto College of Nursing, Ribeirão Preto, São Paulo, Brazil, 2 Department of Nursing, Federal University of Catalão, Catalão, Goiás, Brazil

* fel1pesantos@yahoo.com.br

## Abstract

This study aimed to analyze the discourses of patients who were diagnosed with multidrug-resistant tuberculosis, the perception of why they acquired this health condition and barriers to seeking care in a priority city in Brazil during the COVID-19 pandemic. This was an exploratory qualitative study, which used the theoretical-methodological framework of the Discourse Analysis of French matrix, guided by the Consolidated Criteria for Reporting Qualitative Research. The study was conducted in Ribeirão Preto, São Paulo, Brazil. Seven participants were interviewed who were undergoing treatment at the time of the interview. The analysis of the participants' discourses allowed the emergence of four discursive blocks: (1) impact of the social determinants in the development of multidrug-resistant tuberculosis, (2) barriers to seeking care and difficulties accessing health services, (3) perceptions of the side effects and their impact on multidrug-resistant tuberculosis treatment, and (4) tuberculosis and COVID-19: a necessary dialogue. Through discursive formations, these revealed the determinants of multidrug-resistant tuberculosis. Considering the complexity involved in the dynamics of multidrug-resistant tuberculosis, advancing in terms of equity in health, that is, in reducing unjust differences, is a challenge for public policies, especially at the current moment in Brazil, which is of accentuated economic, political and social crisis. The importance of psychosocial stressors and the lack of social support should also be highlighted as intermediary determinants of health. The study has also shown the situation of COVID-19, which consists of an important barrier for patients seeking care. Many patients reported fear, insecurity and worry with regard to returning to medical appointments, which might contribute to the worsening of tuberculosis in the scenario under study.

such as full name, date of birth, age and local of residence. In addition, data were collected from a small group of participants in a priority city in Brazil. All participants were informed about the study objective, and how their confidentiality would be protected, as well as their rights to withdraw from the study at any time. The participants included in this study gave their consent to use only anonymized quotes in this research. All relevant data are within the paper. The raw individual participant's interview transcripts are not publicly available due to confidentiality and privacy concerns. The first author has registered their details, as well as contact data in case of interest in collaborative work or further information. For more information and data request, contact the Research Ethics Committee at the University of São Paulo at Ribeirão Preto College of Nursing. Phone number: +55 16 3315 9197 and e-mail: cep@eerp.usp.br.

**Funding:** This study was financial in part by National Council for Scientific and Technological Development (CNPq), National Council for Scientific and Technological Development (CNPq) Research Productivity Scholarship and The São Paulo Research Foundation (FAPESP). FLS received financial support from National Council for Scientific and Technological Development (CNPq) Grant 130205/2019-2. Website: https://www.gov.br/cnpq/pt-br. RAA received financial support from National Council for Scientific and Technological Development (CNPq) Research Productivity Scholarship Grant 304483/2018-4 and The São Paulo Research Foundation (FAPESP) Grant 2018/14337-0. Websites: https://www.gov.br/cnpq/pt-br and https://fapesp.br/en. The funders had no role in study design, data collection and analysis, decision to publish, or preparation of the manuscript.

**Competing interests:** The authors have declared that no competing interests exist.

# Introduction

Tuberculosis (TB) persists as a serious public health problem worldwide, despite the advances achieved in controlling the disease in recent years. In addition to the epidemiological situation, multidrug-resistant tuberculosis (MDR-TB) stands out, which has converted over the years from a simple tendency to a risk of a lack of global control of the disease [1].

The incidence of TB is directly related to the increase in poverty, the unequal distribution of income and the accelerated increase in urbanization in large centers. The insufficient control of TB points to the need for effective public health measures. The increasingly recurrent appearance of MDR-TB in the world shows that the therapeutic regimens used are increasingly insufficient due to the different forms of resistance of the disease [2].

Multidrug resistance is considered an evolutionary biological phenomenon that is iatrogenic, as it originates from inadequate treatments, and/or treatment abandonment, due to low potency therapeutic schemes against *Mycobacterium tuberculosis*. When the resistance mainly involves first-line combat measurements (rifampicin and isoniazid), other alternative therapeutic schemes are needed to face resistant strains [3].

According to the World Health Organization (WHO), there were an estimated 465,000 cases of MDR-TB in 2019, equivalent to 11% of the total number of cases of TB worldwide. In the same year, an estimated 3.3% of new TB cases and 18% of previously treated cases had MDR-TB [1]. According to the latest data reported in MDR-TB annual cohorts, of those affected by the disease who started treatment for MDR-TB, only 57% completed the treatment successfully, 16% were lost to follow-up, 7% had treatment that failed, and 15% died, while for 5% there was no outcome information [1]. When these findings are compared with the outcomes in new or recurrent cases of sensitive TB, 85% of cure, they are very low for those with MDR-TB, highlighting one of the main global challenges of the disease [1].

In 2019, the total number of people diagnosed with TB who underwent specific tests for MDR-TB was higher than in previous years. Worldwide, 81% of previously treated patients and 59% of new cases were tested for MDR-TB, which was an increase of 10% and 61%, respectively, compared to 2018. This increase is mainly due to the implementation of the GeneXpert MTB/RIF assay and improved laboratory diagnostics [1].

In Brazil, TB still persists as a major public health problem, having a direct connection with social conditions [3]. In relation to other countries in the world, Brazil is among the 20 countries with the highest burden of TB, being responsible for 0.9% of the estimated cases in the world and 33% of the estimated cases on the American continent [1]. In the state of São Paulo in 2019, the incidence was 50.9/100,000 inhabitants [4], which is a serious public health issue.

Data from a survey conducted in Brazil revealed a predominance of cases of MDR-TB in the acquired form, which indicates that the country still suffers from poor quality and/or inefficient treatments [5]. According to Orenstein and colleagues [6], treatment success occurs in approximately 50% of cases, while deaths and abandonments reach 24%, which are drastically lower values when compared to treatments for sensitive TB.

It is important to highlight the determinants of MDR-TB due to the consequences of this form of disease on people with TB, who are subjected to expensive (to both the health system and patients), extensive, complex, and stigmatizing treatments and the use of drugs with high toxic potential that consequently lead to a worse prognosis and exorbitant investments in the public health system [7, 8].

According to the WHO, people affected by MDR-TB spend up to 20 times more compared to people affected by drug-sensitive TB [9]; it is worth mentioning that the investment of families with treatment is generally above 20% of the annual family income, which has an impact on families with less purchasing power [10].

The literature has evidenced the main determinants related to MDR-TB, such as individual characteristics (sex, age, educational level), behavioral issues (smoking, alcohol, substance abuse), clinical factors (HIV, diabetes mellitus, systemic arterial hypertension or other associated disease), a history of previous treatment, type resistance, access to quality and resolute health services, social protection, compensatory policies, income transfer and improvements in the social context [11].

In addition, the poor educational degree has an important role in the onset and in the MDR-TB treatment loss to follow-up. The low educational level, as well as low-income, is associated with poor infrastructures, limiting the access to health services. Some behavioral issues, such as alcohol abuse, are also predictors that determine the treatment successfully or treatment failure outcomes. The low-income groups tend to have higher levels of related to behavioral issues harm than high-income groups [12, 13].

Social protection interventions are relevant strategies that support to reduce poverty, and consequently, provide better treatment outcomes, with additional effects in reducing MDR-TB cases [12, 13]. These determinants have a great influence on patients seeking care as well their adherence to treatments and prescription regimes [14].

Studies evidenced the factors associated with non-adherence such as the long duration of treatment, drug intolerance, lack of knowledge about the disease, poverty and inequality in locations where patients live, and weak support from healthcare workers [15], as well as adherence; the reception performed within the health services is configured as an important factor the patient's compliance in the therapeutic scheme [16].

Although studies have been carried out to understand the factors leading to patients developing acquired TB-MDR, most of them were developed in a quantitative approach [16, 17]; few studies included the perception and assessment of patients and their experiences in seeking care, as well as the associated challenges [18], mainly in Brazil.

Patient perception is equally relevant for subsidizing health policies and qualifying care, avoiding new episodes and spreading drug-resistant TB strains, which are more aggressive, severe and difficult to treat and control.

On March 11 2020, the World Health Organization (WHO) announced COVID-19 as a pandemic, registering, at the time, 118,319 cases and 4,292 deaths on different continents. In less than a month, the number of cases of infection by the SARS-CoV-2 virus, the causative agent of COVID-19, reached 1,279,722, with 72,614 deaths directly related to the infection worldwide [19].

Unlike COVID-19, TB has been known to affect humanity for thousands of years and has been declared a global emergency by the WHO since 1993. According to the WHO, an estimated 10 million people are affected by TB worldwide, with more than 1.2 million people dying each year and 3,014 dying per day [1].

Although there are still no studies associating the worst clinical outcomes of COVID-19 with TB, it is important to consider that both *Mycobacterium tuberculosis* and SARS-CoV-2 mainly attack the lungs, and affect the most vulnerable, both biologically and socially.

Therefore, the aim of this study was to analyze the discourses of patients who were diagnosed with MDR-TB, the perception of why they acquired this health condition and barriers to seeking care in a priority city in Brazil during the COVID-19 pandemic.

## Materials and methods

### Study design

This was an exploratory qualitative study [20], which used the theoretical-methodological framework of the Discourse Analysis (DA) of French matrix [21], guided by the Consolidated Criteria for Reporting Qualitative Research (COREQ) [22].

## Theoretical-methodological framework

DA moves through different areas of knowledge; from this perspective, it articulates linguistics, historical materialism and psychoanalysis. Thus, DA would be the intermediary, since it is formed in the place where language has to be referred to its exteriority. With these considerations, it is worth reflecting that DA seeks a specificity that it dismembers in ideology and discourse [23].

To understand the theoretical-methodological orientation of DA and to analyze the data, it is necessary to consider the key concepts that constitute its theoretical basis: subject, ideology and discourse [24].

The individual is the result of the interaction of several voices, and of the relationship with the socio-ideological; therefore, it has a heterogeneous character. For DA, the subject is essentially ideological and historical, as it is inserted in a certain place and time [25].

Ideology is defined as a concept of the world of the person enrolled in a given social group in a historical circumstance [26]. Therefore, ideology is a fundamental concept for DA, since it combines the linguistic with the socio-historical [23].

For DA, discourse is defined as a set of statements regulated in the same discursive formation. It is with language that the subject is constituted and it is also in language that he leaves the marks of this ideological process [27]. Discourse is the point of articulation of ideological processes and linguistic phenomena. The language as interaction is a mode of social production, which is neither neutral nor natural, being the privileged place for the manifestation of ideology, that is, of the ideological formations that are directly linked to the subjects. It could be said that one does not start from ideology to meaning, but seeks to understand the effects of meaning from the fact that it is in the discourse that the relationship of language with ideology is configured [24].

It is understood that discourse is the expression of opinions, attitudes, speeches and representations of the subject, denoting an outline of a certain moment, aimed at a process of analysis of the senses [24, 28].

Therefore, it can be inferred that DA seeks to understand language, as a symbolic work originating from historical materialism, starting with a general social work of the subject, constitutive of man and his history [24, 29, 30].

## Study setting

The study was carried out in Ribeirão Preto, an inland city of the state of São Paulo (SP), Brazil. Located 314 kilometers (km) from the state capital, Ribeirão Preto has an area of approximately 650 square kilometers ($km^2$) and a high population density of 995.3 inhabitants/$km^2$. It also had an estimated population of 711,825 inhabitants in 2020, of which 99.7% live in urban areas [31] (Fig 1).

The healthcare network of Ribeirão Preto is divided into five Health Districts: North, South, East, West and Central, totaling 49 establishments of Primary Health Care, five of which are Basic District Health Units, 18 Family Health Units and 26 Basic Health Units [32, 33]. As for the hospital network, it has 18 units, including three public hospitals, a university hospital, seven philanthropic hospitals and eight private hospitals [32, 33].

All of the Basic Health Units can and should activate case identification for respiratory symptoms, with sputum smear microscopy and/or X-ray request. However, despite the fact that TB is considered a treatable disease in the primary healthcare services in Ribeirão Preto, treatment and follow-up with specialized infectious disease clinics in each of the five districts was chosen [32].

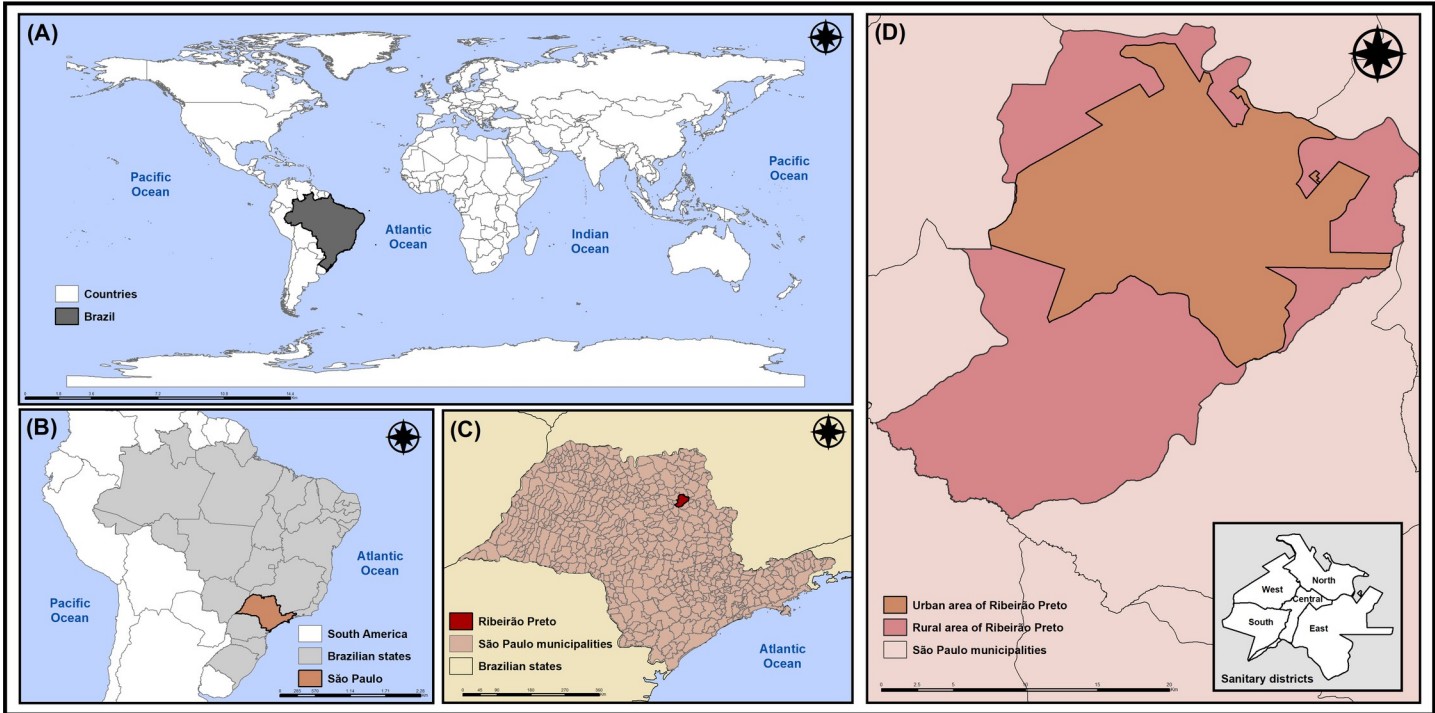

**Fig 1. Geographic location of Ribeirão Preto, São Paulo, Brazil.** (A) Brazil; (B) State of São Paulo; (C) City of Ribeirão Preto; (D) Sanitary Districts of Ribeirão Preto. Source: Authors.

It is noteworthy that the outpatient clinic of the Clinics Hospital of the Ribeirão Preto Medical School at University of São Paulo (HCFMRP-USP) is the reference for patients with MDR-TB and also accompanies patients with TB-HIV co-infection [32, 33].

## Participants

We have recruited men and women over 18 years of age who are undergoing treatment for MDR-TB in the city of Ribeirão Preto. The interviews were conducted with the participants who consented to attend to the study. It is important to highlight that the participants had no physical or psychological limitations that impeded them from participating in the study.

## Sampling, sample size and non-participation

Participants were recruited by purposive-convenience sampling from a reference university hospital specialized in the treatment of people affected by MDR-TB in Ribeirão Preto.

Potential participants were identified in the university hospital's database, with the initiation of treatment occurring between January 2019 and June 2020. Initially, participants were given information about the study and were subsequently invited to participate. The first contact with the possible study participants was achieved by telephone call to present the research objective. Those who accepted the invitation were given new appointments for the interviews.

In total, fourteen patients were invited to participate of the study, with seven agreeing to participate. Four potential participants did not agree to participate in the study, two did not have the correct telephone number in the university hospital's database and one did not answer the telephone call on the scheduled day of the interview.

The interviews were guided by the concepts of horizontal exhaustiveness and vertical exhaustiveness [24].

For DA, saturation is not aimed at; this is also known as horizontal exhaustiveness, that is, neither extension nor completeness, or exhaustiveness in relation to the empirical object. By definition, every discourse is established in relation to a previous discourse and points to another [24].

The aim is vertical exhaustiveness, which is considered in relation to the objects of the analysis and its theme. In this way, vertical exhaustiveness leads to relevant theoretical consequences and does not treat the data as mere illustrations. The data are analyzed as facts of the language with its memory, semantic thickness, and linguistic-discursive materiality [24].

## Interviews

Semi-structured interviews were conducted based on the theoretical framework of Social Determinants of Health [34]. This data collection technique was adopted since it is the most recommended approach to build participants' discourses centered on their experience [35].

In 2010, the WHO proposed a conceptual framework for action on the Social Determinants of Health, the aim of which is to highlight differences between the levels of causality and to distinguish the mechanisms by which social hierarchies are created [34].

In general, the conceptual framework of the WHO is based on two types of Social Determinants of Health: structural determinants and intermediary determinants. Structural determinants affect equity in health and well-being, through their action on intermediary determinants. From this, a differentiation emerges between structural determinants of health inequities and intermediary determinants of health, which together constitute the Social Determinants of Health [34].

Structural determinants include all of the social and political mechanisms that generate, shape and maintain social hierarchies, including the labor market, the educational system, political institutions and other cultural and social values. These determinants cause stratification and social class divisions in society, and define the individual socioeconomic position within the hierarchies of power, prestige and access to resources. Structural mechanisms are rooted in the main institutions and processes in the socioeconomic and political context [34].

Intermediary determinants are represented for elements categorized into behavioral and biological factors, material circumstances (living and working conditions, availability of food, housing, etc.) and psychosocial factors. This set of determinants also includes the health system and social cohesion [34].

The interviews were conducted after reading and accepting participation in the research through the Informed Consent Form of each participant. Those who agreed to participate verbally consented. The data were collected by telephone calls, considering the necessary measures of social distancing during the SARS-CoV-2 pandemic, in the period between June and August 2020.

Each interview lasted for an average of 55 minutes, was conducted by the main researcher and was recorded using an electronic recorder for full transcription at a later date.

The semi-structured interview script was developed based on an extensive literature review on the determinants of TB and MDR-TB, as well as the approach of health professionals and the health services to people affected with TB, the COVID-19 pandemic impact during the treatment and the possible adaptations necessary during treatment.

The first author documented notes about his experiences during interviews and immediately after each one; these notes were considered reflections and preliminary data analytical ideas [36]. The interviewing author was a male nurse with experience in conducting qualitative interviews, and who did not belong to the staff where the research was conducted.

## Data analysis

The Atlas.ti 8.0 software (ATLAS.ti Scientific Software Development GmbH) was used to organize and assist in the analysis of empirical data, which helped with the treatment of the material produced in an organized and systematic way. The interviews were inserted into the software to create a Hermeneutic Unit. After reading these, the discursive fragments were selected in order to determine the elements that integrate the object of study, through which the codes originated (from what emerges in the discourses). Thus, the corpus of the study was constituted in a set of discourses, assisting in the identification of traces and evidence, reconstructing the historical memory of the interviewed participants.

To guarantee the anonymity of the research participants, in the presentation of the discursive excerpts, the letter "P" was used to indicate the participant, followed by a sequential number from one to seven, according to the order of the interviews.

The empirical data were analyzed from the perspective of DA, where it is proposed to pass the raw material, such as the transcribed interviews, to the discursive object, through the following steps: (1) from the linguistic surface to the discursive object: we work with the production conditions, which comprise the subjects and the situation. In other words, it goes from the circumstances of the enunciation to a broader way: the socio-historical and ideological context. What has been said, the way it was said, who said it and the specific discursive circumstances were observed; (2) from the discursive object to the discursive process: this relates to the distinct discursive processes and to the ideological formation that governs these. It is at this moment that the constitution of the discursive processes is reached, which will be responsible for the effects of meanings produced in the initial empirical and symbolic material; (3) discursive process (ideological formation): the analytical phase, notably marked by the relations of discursive formations with the effects of ideology. Thus, through ideological formations, linguistic-discursive marks will be observed in the discourses of the participants who comprised the study [24].

In this sense, DA shows the constant movement from linguistic materiality to the exterior of the socio-historical and ideological context, characterized according to a space of production and the movement of discourses and meanings [24].

## Ethical aspects

The study was approved by the Research Ethics Committee at the University of São Paulo at Ribeirão Preto College of Nursing under Certificate of Presentation for Ethical Consideration number 19520819.0.0000.5393 and Report number 3.654.833, issued on October 22nd 2019 in accordance with the Guidelines and Regulatory Standards for Research with Human Subjects, Resolution number 466/2012 of the National Health Council of Brazilian Ministry of Health.

An informed consent form was required for each participant. All participants were informed about the study objective, and how their confidentiality would be protected, as well as their rights to withdraw from the study at any time. Those who agreed to participate verbally consented.

The participants included in this research gave their consent to use only anonymized quotes in this study.

## Results

Seven participants were interviewed; up to the time of the interview, these individuals underwent treatment at the university hospital. The seven interviews totaled 385 minutes of interviews.

Table 1 presents the socio-demographic, clinical and operational characteristics for each study participant, defined in the theoretical-methodological framework of DA as conditions of discourse production.

**Table 1. Socio-demographic, clinical and operational characteristics of participants included in the study.**

| Characteristics | N | Percentage (%) |
|---|---|---|
| Socio-demographic | | |
| **Sex** | | |
| Male | 3 | 43% |
| Female | 4 | 57% |
| **Age** | | |
| 18–30 years | 1 | 14% |
| 31–59 years | 5 | 72% |
| ≥ 60 years | 1 | 14% |
| **Self-identified ethnicity** | | |
| White | 6 | 86% |
| Brown | 1 | 14% |
| **Education** | | |
| Primary | 4 | 57% |
| High school | - | - |
| Complete undergraduate | 2 | 29% |
| Complete graduate | 1 | 14% |
| **Profession/ Employment** | | |
| Unemployed | 3 | 43% |
| Formal Work | 1 | 14% |
| Student | 1 | 14% |
| Pensioners | 2 | 29% |
| **Beneficiary cash transfer programs** | | |
| No | 7 | 100% |
| Yes | - | - |
| Clinical-operational | | |
| **Type of case** | | |
| New | 4 | 57% |
| Retreatment | 2 | 29% |
| Relapse | 1 | 14% |
| **MDR-TB HIV co-infection** | | |
| No | 7 | 100% |
| Yes | - | - |
| **MDR-TB diabetes comorbidity** | | |
| No | 5 | 71% |
| Yes | 2 | 29% |
| **Alcoholism** | | |
| No | 6 | 86% |
| Yes | 1 | 14% |
| **Drug addiction** | | |
| No | 6 | 86% |
| Yes | 1 | 14% |
| **Smoking** | | |
| No | 6 | 86% |
| Yes | 1 | 14% |
| **Treatment side effects** | | |
| No | 2 | 29% |
| Yes | 5 | 71% |

(*Continued*)

**Table 1.**  (Continued)

| Characteristics | N | Percentage (%) |
|---|---|---|
| **Clinical form** | | |
| Pulmonary | 7 | 100% |

The analysis of the participants' discourses allowed the emergence of four discursive blocks, which, through discursive formations, revealed the determinants of MDR-TB.

## Impact of the social determinants in the development of multidrug-resistant tuberculosis

The socioeconomic profile is one of the determinants that has an intimate and direct relationship with TB. Regarding drug resistance, it is evident from the participants' discourses that this marker of social inequality still persists as a strong determinant for MDR-TB.

> *"My son got sick first, but my house is not big, and we all sleep in the same room, I believe that's why I got sick too"* (P. 01)

> *"Ah, during my first treatment, I lost my job, and I was losing a lot of weight. I work as a cleaner. . . As I was very weak, I was without work and I was unable to buy food for home. I was having help from my neighbors. I think that's why the first treatment didn't work. . ."* (P. 04)

> *"When I started feeling the first symptoms, I didn't want to believe that I was starting to get sick! I am the only person in my house who works. . . when I was diagnosed with tuberculosis I spent the first few months without working. . . the drugs made me very bad. . . the doctor said that I needed to eat well to start gaining weight, but how could I eat well if I didn't have the enough money to buy food for my house?"* (P. 05)

> *"After six months he (son) had another one, then he took the exam and was diagnosed as having multidrug resistance. It was shocking for us who live. . . all together, so it was really hard!"* (P. 07)

Four of the research participants reported that they receive between 1 and 2 minimum wages.

> *"Look, last year the retired came. I get a minimum wage."* (P. 01)

> *"It's just a minimum wage."* (P. 02)

> *"The income? It is 2,000 Brazilian Reals per month for 5 people living in my house."* (P. 05)

Still in relation to the discursive block, the participants' positions are inscribed in a way that focuses on the basic food baskets and income as a facilitator for the treatment of MDR-TB.

> *"Ah, a salary. . . I received a food package there, but since they cut it there, there is no more. . . Ah, help, right. Money is not enough. If I received 4, 5 thousand or 6 thousand I wouldn't even need. . . Yeah, yeah. . . yeah. . . there are people who don't get any money, they don't get anything, then they're having a hard time, you know, because it was. . . this food basket."* (P. 04)

## Barriers to seeking care and difficulties accessing health services

The relationship between healthcare providers and users of health services, the organizational structure, and the availability of vehicles, drivers and medication are also elements perceived

by the research participants as barriers to accessing health services and the continuity of MDR-TB treatment.

> *"Look, like this, for me it is difficult because it is very far, you know, far from my home. You have to get up like four o'clock in the morning to get there at 7:30 am or 8 am at the Hospital. So, like this, it's a little difficult, but I have to. . . have to go to the doctor to be able to get medication, right. Sometimes I, I, I ask to give me medication for 60 days, right. But then it is more complicated, right, because it becomes very tiring for me, right."* (P. 02)

> *"To the Hospital of Clinics I go. . . I go with a city hall car, right."* (P. 02)

Health services, especially those that are a reference for the treatment of MDR-TB, have a fundamental role in forming bonds between users of health services for adherence to treatment and the consequent achievement of positive outcomes.

> *"Ah, I have no complaints, they treated me well there, all the times I had there, I didn't have it like that. . . no complaints. Why is there a place that. . . they spoke already, I know. . . I know the case of people who stayed from hospital to hospital."* (P. 04)

> *"We are always very well treated, right, at the time of treatment, they are very careful with us, they treat us very well, right."* (P. 05)

> *"The hospital gave me a lot of attention, all the support, they were very affectionate, both in the hospital and in the outpatient treatment, they were always attentive, careful, caring, you know, I don't have to say, ready to solve my situation, always talking to the boss team bringing the recipe stuff. . ."* (P. 07)

## Perceptions of the side effects and their impact on multidrug-resistant tuberculosis treatment

The drugs used in the treatment of MDR-TB and the long period of exposure influence the appearance of side effects that affect the dynamics of adherence to treatment. It was observed that the research participants had side effects from the first few days of treatment.

> *"Ah, at the beginning it was, you know, some red pills for tuberculosis, you know, I wanted to vomit the food, it looks like it wouldn't go down or up, it wouldn't come out. . . I couldn't vomit afterwards."* (P. 04)

> *"A lot of dizziness and severe pain in the chest, right."* (P. 02)

> *"A little bit of myalgia. Arthralgia in the first treatment."* (P. 06)

> *"Ah, when the treatment was here. . . I didn't gain weight and I was in a lot of pain."* (P. 03)

The high neurotoxic and hepatotoxic potential of the drugs used in the treatment of MDR-TB requires special attention from managers and health professionals. However, despite the high toxicity potential, the prescription of these drugs is part of the current protocols for the management of MDR-TB.

> *"Look, I have some numbness in my legs, you know, from the medicine. Even tomorrow, I have another return. . . It looks like they are going to decrease the medication already. They will decrease the dose, right. . ."* (P. 01)

*"It was awful. I felt it was terrible, horrible, feeling unwell, I felt like retching, but then talking to the doctor, they changed the medication there. . . everything went well."* (P. 05)

The side effects from medications need to be taken into account in the treatment adherence process by patients. Patient-centered care is an important modality that can favor not only treatment adherence, but also the best outcomes for MDR-TB.

*"No, I didn't think about giving up. It was bad, it sucked. That malaise, that bad thing in your head, you know, a lot of retching when I took the medication, but then I said no, I need to be strong, I need to be strong."* (P. 05)

*"Then I talked to the doctor, then he said. . . so, we will then change your medication."* (P. 02)

*"It has effects, but it has this side that needs to be studied more carefully, right, the side effects. I believe that it is not just Ethambutol. . . So I think it would be a good idea to focus a little more on these effects of all medications, you know."* (P. 07)

*"Ah I thought so, oh, what for? I don't have. . . that lung is already lost. Why do I have to sacrifice myself going there, take these medications and it's not working? Huh. But then, I think like that, no. . . if I'm going, if you're asking me to go, something is better, right. But then I took it out of my mind, then I already have something else to think about, right, that I have to continue to see what happens, right."* (P. 06)

*" In this sense, the nurse is very friendly, a counselor, she advised us not to give up, not to be discouraged, you know, to go ahead that everything would be all right.."* (P. 04)

## Tuberculosis and COVID-19: A necessary dialogue

The COVID-19 pandemic unveiled health inequities, especially in relation to diseases that are closely related to socioeconomic conditions.

The request for information arose from the perceived susceptibility to COVID-19 and its perceived severity "because people are dying". Participants expressed a high level of fear and anxiety about the possibility of infection by the new coronavirus because there is no known treatment, it requires prolonged periods of isolation from social life and, in its most severe form, can be fatal.

*"So I stayed in April and May without going there and I was rescheduling and taking the medicine so I could wait a little bit, right. Why, yeah, the fear, I did, right. . ."* (P. 01)

*"Leaving the house, being able to go to the doctor, fear, a feeling of fear. . . Catching another disease and knowing that I'm being treated for a lung disease, right?"* (P. 02)

*"I was more afraid of COVID. . . because it is such a fast disease, you know."* (P. 04)

*"Look, like this, I try not to leave the house, I'm afraid to go out to get the medicine. I call the hospital when I have to get the medication, you know, they receive the prescription and send it to me, so I can stay isolated. . ."* (P. 02)

However, some participants claim that the MDR-TB and COVID-19 co-infection was a lethal combination, given the severity of the two diseases and the weight that COVID-19 could represent the already tense health system.

*"Oh, like, because he said that anyone who has a problem with a chronic disease would get killed. But so far I haven't caught it. . ."* (P. 03)

*"Now comes the Coronavirus, I was really scared, so why? Because of my lung problem, you know, I thought so. . . my God! I have to take all the care in the world, because if I get a disease like this, I'm dead. So, I was a little scared, but I took it very well, I'm following all the rules, I'm not without a mask, since when I'm in the treatment there, I shoot just to sleep because it's suffocating, but everywhere it's alcohol gel, it is alcohol, I take all these precautions because I am scared to death with this problem that I have."* (P. 05)

*"Yeah. . . we were more afraid, you know, more worried, I think I'm taking care of myself, more. . . than if I hadn't been treating, but. . . I'm still working, so be careful."* (P. 06)

Participants also described how social distancing measures have negatively affected their regular social activities, restricting meetings with friends and loved ones on special occasions and preventing them from carrying out their daily activities. Although many expressed a fear of contracting COVID-19 during their routine consultations, they sought help for both MDR-TB and COVID-19.

*"Wow, it was complicated. . . Look, like this. . . we are a little afraid to even try to avoid it. . . talking to people like that, because when we do the treatment we end up meeting many and then you always look for to avoid getting too close, you know, to avoid that I think my only fear was that someone in my family could get it, you know, the fear of someone in my family getting it, right?"* (P. 07)

*"Like this, I stay inside my house a lot. . . But every now and then I feel like going to my mom's house. . . but I know that it is complicated to visit her! In addition to COVID, I am afraid that she will also become ill due to tuberculosis"* (P. 05)

## Discussion

The study aimed to analyze the discourses of patients who were diagnosed with MDR-TB, the perception of why they acquired this health condition and barriers to seeking care in a priority city in Brazil during the COVID-19 pandemic. The study showed that the structural determinants of health inequities, such as the socioeconomic and political context, and the intermediary determinants of health, have strong influences on the development of MDR-TB.

In the impact of the social determinants in the development of MDR-TB block, the study corroborates that which has been reported in the literature in the last decades of the strong influence of socioeconomic conditions as risk factors for TB and for MDR-TB [37–41].

The conditions of poverty and extreme poverty affect vulnerable groups who are less able to enter the labor market [42]. According to the World Bank criteria, defining conditions of extreme poverty and poverty, Brazil has 13.5 million people, an approximate percentage of 6.5% in a state of extreme poverty (monthly income per capita less than 145 Brazilian Reals) and 52.5 million, about 18.7%, in a state of poverty (living with monthly per capita income less than 420 Brazilian Reals) [43]. Another issue is that 29.5 million people in the country are in a state of poverty and do not have access to a sanitation system, 13.5 million are not served by a piped water supply, and 11.1 million do not have garbage collection services. Income defines the proportion of school dropouts and delays for young people aged 15 to 17. About 11.8% of poor young people drop out of school during high school, and 33.6% have delays in their schooling [42].

In a study carried out in the Southeast region of Brazil, the offer of a basic food basket and transport voucher were the indicators evaluated with the worst averages, with no statistically significant difference [44]. A study was conducted in five priority municipalities for TB control

in the Southeast and Northeast regions, which included the city of Ribeirão Preto; this study demonstrated that the worst averages in relation to access to TB treatment are related to the lack of provision of transportation vouchers by health services. However, the provision of basic food baskets was assessed as satisfactory in these same services [45].

When the influence of socioeconomic factors on TB treatment adherence is recognized, it is clear that, even when treatment is free, the absence of benefits or support to cover expenses with transportation and food can hinder the continuity of treatment [45]. However, the provision of this type of support requires health services to be more integrated with other social assistance equipment and exemplifies the interrelationship of organizational and economic accessibility.

In the access to health services discourse block, the research participants build the therapeutic itinerary of access to health services based on the reception received at the health units. In addition, how the relationship between health professionals and users of health services influences the positive outcomes of MDR-TB treatment is discussed.

It is widely known that barriers related to distance are associated with the level of complexity of services: the less specialized they are, the closer to the population they must be, and vice versa. In this sense, studies have found that access to health services is criticized by users, when the distance was not short enough and was not in the territory of the user's home [46].

A qualitative study that sought to analyze the reasons that lead TB-HIV co-infected patients to abandon TB treatment found that the factors related to patients who contribute to abandonment are low socioeconomic status, adverse effects of medications, drug use and little personal motivation. In addition, factors related to health services that also contribute to treatment abandonment were identified, such as physical structure, organization of the work process and access [47].

Regarding the bond, the results show that TB patients are often treated by the same professional, who provides enough time for the patient to raise any concerns and always understands and clearly answers the doubts presented.

The fact that people affected by MDR-TB are cared for by the same professional presupposes an approximation, so that patients have, in these professionals, a reference in their healthcare. Similar results were found in surveys carried out in India, Russia and Uganda that showed greater bonding when professionals know and are known by users who seek them out in the face of a new event, or to raise concerns [48–50].

Study participants suggested a patient-centered model of care which includes appropriate treatment and a support group for MDR-TB patients to ensure the successful completion of treatment.

In the third discourse block, perceptions of the side effects and their impact on multidrug-resistant tuberculosis treatment, the participants build their discourses based on the process of treating MDR-TB and how the side effects influence the process of obtaining a cure.

The presence of unpleasant bodily sensations constantly remind research participants that they still have a problem with MDR-TB. Some individuals, when the symptoms improve, believe that they are no longer sick, since the concept of the disease is associated with the presence of undesirable symptoms that do not allow them to act as before. Taking drugs to relieve a disease that is no longer physically manifest seems counterintuitive [51].

The effects of drugs and reactions to treatment involve issues that are equally important for adherence. According to Walker and colleagues [52], these issues are the expectations of treatment and cure, motivation for maintaining health, and the personal advantages of continuing with therapy, in addition to the benefits of the treatment regimen that demarcate some behaviors during treatment. Some sick individuals find the drugs strong enough to negatively affect other parts of their body; both concepts can lead to non-adherence [53].

MDR-TB has an impact on affected individuals and allows comprising the meaning of becoming ill through their perceptions and how the subjective weight that the disease has for each individual seems to be related to moments of tension, anxiety and fear during the treatment until the cure. The disease exposes affected individuals as fragile, affecting the way of living, changing the established order, and changing their relationships with other people [54].

In the last discursive block, tuberculosis and COVID-19: a necessary dialogue, the research participants reported their discourses traversed by the Sars-CoV-2 pandemic that started in the first months of 2020. In August 2020, 75% of TB programs in 106 countries reported service interruptions due to COVID-19 [55], highlighting the need to improve access to TB care during this public health emergency [56, 57].

The pressure on health services resulting from the COVID-19 pandemic, combined with impacts on care-seeking behavior, could slow or reverse progress towards TB treatment and prevention targets set at the United Nation high-level meeting on TB, especially in high TB burden countries [1].

As the pandemic has become the focus of global attention, it should not be seen as merely a phenomenon of scientific interest until it is controlled. The pandemic must be seen in light of the main socio-economic impacts and their disruptive effects on routine health services, as well as in progress towards sustainable development goals (SDGs). A modeling analysis commissioned by the Stop TB Partnership concluded that the global response to the COVID-19 pandemic is likely to have drastic and damaging consequences for TB services [58].

In the first two months of 2020 was identified reductions in TB-related hospital discharges, newly diagnosed cases of active TB, total active TB outpatient visits, and new latent tuberculosis infection (LTBI) cases and LTBI outpatient visits. These results may be explained by a general decrease in the use of health services [59].

Resources for TB service provision were reassigned to other medical services. Outpatient visit numbers may have decreased because of patients' fear of exposure to COVID-19 [60]. Access to health services may have decreased because of interruptions in or difficulty accessing public transportation, although health-related travel was permitted in most countries [61]. There is already evidence from several high TB burden countries of large reductions in the monthly number of people with TB being detected and officially reported in 2020 [1].

Governments of countries with a high burden of TB and MDR-TB need to ensure the continuity of services during the COVID-19 pandemic. This includes being proactive in protecting the most vulnerable, including protection from economic hardship, isolation, stigma and discrimination. The global response needs to identify and mitigate potential risks to the critical mission of combating TB as well as other respiratory diseases. This includes identifying how capacities and infrastructures can be adapted to strengthen the response to COVID-19 and allow flexibility in programs to enable countries to respond as COVID-19 evolves. All stakeholders with an interest in TB must remain closely engaged with the WHO to ensure close coordination in response to COVID-19 and to sustain progress towards the Sustainable Development Goals (SDG) of health and well-being for all [57].

As the COVID-19 pandemic spreads to high-TB regions, countries must implement strategies to alleviate pressure on healthcare systems and mitigate disruption to routine health services [62]. Current measures of social distancing and lockdown make it particularly challenging for TB programs to provide the diagnosis, treatment and care for communities affected by TB. To ensure that there are no interruptions in healthcare services, TB programs will have to identify and rely on alternative options, such as virtual care, digital health and community monitoring solutions to bring the necessary services as close as possible to the people and communities affected by the disease [63, 64].

It is crucial to maintain and strengthen TB services as an essential component of overall progress towards universal health coverage and resilient health systems, and to ensure synergies in the responses to both TB and COVID-19 [1].

As a limitation of the study, the number of patients registered for treatment was limited, which was reflected by the sample used, as that was also limited. Although the results of the study cannot be generalized, this study provided an opportunity for patients to be understood with regard to the determinants and the successful completion of MDR-TB treatment. This can help program managers and service providers to understand the motivators and design strategies to improve success rates.

The study contributes to the advancement in knowledge since the most of studies about TB-MDR have emphasized biological, clinics and epidemiological aspects, as well as the perception of health professionals; few studies have included the patients' perspective, which is valuable for defining action plans and policies addressed to face the issue. The study has also shown the situation of COVID-19, which consists of an important barrier for patients seeking care. Many patients reported fear, insecurity and worry about returning to medical appointments which might contribute to the worsening of TB in the scenario under investigation.

## Conclusions

Considering the complexity involved in the dynamics of MDR-TB, advancing in terms of equity in health, that is, in reducing unjust differences, is a challenge for public policies, especially at the current moment in Brazil, which is experiencing an accentuated economic, political and social crisis.

The importance of psychosocial stressors and the lack of social support should also be highlighted as intermediary determinants of health. In this context, the health system also plays an important role mainly because of its influence on access barriers. Finally, social cohesion is proposed as a transcendental determinant and is understood as a way in which to account for the impact of relationships established between communities and institutions, particularly the State, on the quality of life of people affected by MDR-TB.

## Supporting information

**S1 File. Semi-structured interview script.**
(PDF)

**S2 File. Consolidated Criteria for Reporting Qualitative Research (COREQ): A 32-item checklist for interviews and focus groups.**
(PDF)

## Acknowledgments

The authors would like to thank the Epidemiological Surveillance Center Professor Alexandre Vranjac, Secretary of Health of the State of São Paulo and Clinics Hospital of the Ribeirão Preto Medical School at University of São Paulo (HCFMRP-USP) for making the data available.

We would also like to thank the seven participants who took time out of their busy schedules to participate in this study.

## Author Contributions

**Conceptualization:** Felipe Lima dos Santos.

**Formal analysis:** Felipe Lima dos Santos, Ana Carolina Scarpel Moncaio.

**Funding acquisition:** Ricardo Alexandre Arcêncio.

**Investigation:** Felipe Lima dos Santos, Ludmilla Leidianne Limirio Souza, Alexandre Tadashi Inomata Bruce, Juliane de Almeida Crispim, Luiz Henrique Arroyo, Antônio Carlos Vieira Ramos, Thaís Zamboni Berra, Yan Mathias Alves, Alessandro Rolim Scholze, Fernanda Bruzadelli Paulino da Costa, José Francisco Martoreli Júnior, Ana Carolina Scarpel Moncaio, Ione Carvalho Pinto, Ricardo Alexandre Arcêncio.

**Methodology:** Felipe Lima dos Santos, Ana Carolina Scarpel Moncaio.

**Project administration:** Felipe Lima dos Santos.

**Supervision:** Ana Carolina Scarpel Moncaio.

**Validation:** Ana Carolina Scarpel Moncaio.

**Writing – original draft:** Felipe Lima dos Santos, Ludmilla Leidianne Limirio Souza, Alexandre Tadashi Inomata Bruce, Juliane de Almeida Crispim, Luiz Henrique Arroyo, Antônio Carlos Vieira Ramos, Thaís Zamboni Berra, Yan Mathias Alves, Alessandro Rolim Scholze, Fernanda Bruzadelli Paulino da Costa, José Francisco Martoreli Júnior, Ana Carolina Scarpel Moncaio, Ione Carvalho Pinto, Ricardo Alexandre Arcêncio.

**Writing – review & editing:** Felipe Lima dos Santos, Ana Carolina Scarpel Moncaio, Ricardo Alexandre Arcêncio.

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
