## [Decision Letter · Decision Letter 0]

15 Mar 2021

PONE-D-21-05519

Patients’ perceptions regarding multidrug-resistant tuberculosis and barriers to seeking care in a priority city in Brazil during COVID-19 pandemic: A qualitative study

PLOS ONE

Dear Dr. Felipe Lima dos Santos,

Thank you for submitting your manuscript to PLOS ONE. After careful consideration, we feel that it has merit but does not fully meet PLOS ONE’s publication criteria as it currently stands. Therefore, we invite you to submit a revised version of the manuscript that addresses the points raised during the review process.

We look forward to receiving your revised manuscript.

Kind regards,

Francesco Di Gennaro

Academic Editor

PLOS ONE

Journal Requirements:

2. Please upload the interview guide used as a Supplementary file.

4. Please amend the manuscript submission data (via Edit Submission) to include author Yan Mathias Alves.

5. We note that Figure 1 in your submission contains map images which may be copyrighted. All PLOS content is published under the Creative Commons Attribution License (CC BY 4.0), which means that the manuscript, images, and Supporting Information files will be freely available online, and any third party is permitted to access, download, copy, distribute, and use these materials in any way, even commercially, with proper attribution. For these reasons, we cannot publish previously copyrighted maps or satellite images created using proprietary data, such as Google software (Google Maps, Street View, and Earth). For more information, see our copyright guidelines: http://journals.plos.org/plosone/s/licenses-and-copyright.

A) You may seek permission from the original copyright holder of Figure 1 to publish the content specifically under the CC BY 4.0 license. 

B) If you are unable to obtain permission from the original copyright holder to publish these figures under the CC BY 4.0 license or if the copyright holder’s requirements are incompatible with the CC BY 4.0 license, please either i) remove the figure or ii) supply a replacement figure that complies with the CC BY 4.0 license. Please check copyright information on all replacement figures and update the figure caption with source information. If applicable, please specify in the figure caption text when a figure is similar but not identical to the original image and is therefore for illustrative purposes only.

Additional Editor Comments:

dear authors follow reviewer suggestion to improve your paper

Reviewers' comments:

Reviewer's Responses to Questions

**Comments to the Author**

1. Is the manuscript technically sound, and do the data support the conclusions?

Reviewer #1: Partly

Reviewer #2: Yes

2. Has the statistical analysis been performed appropriately and rigorously? 

Reviewer #1: No

Reviewer #2: Yes

3. Have the authors made all data underlying the findings in their manuscript fully available?

Reviewer #1: No

Reviewer #2: Yes

4. Is the manuscript presented in an intelligible fashion and written in standard English?

Reviewer #1: No

Reviewer #2: Yes

5. Review Comments to the Author

Reviewer #1: The aim of the study was to analyse the discourses of participants who were diagnosed with MDR-TB, the perception of why they acquired this health condition and barriers to seeking care during the Covid-19 pandemic. I tried to see whether the objectives have been met.

Of the four discursive blocks formed, the first; “why did the disease develop? Social and economic factors” is expected to bring out the perceptions of the participants on the causal factors of MDRTB. Though the discursive analysis did bring out the fact that the participants have a low socioeconomic status, the verbatim did not suggest that the participants have perception that their low socioeconomic condition has causal relationship with their disease condition. The discursive analysis did not yield significant additional information than what could have been inferred from the quantitative data on the sociodemographic profile as four among the seven are unemployed.

The second discursive block “barriers to seeking care and difficulties assessing health services” has met the objective.

The third discursive block “a blessing in disguise: side effects of multi drug resistant tuberculosis treatment” has many weaknesses. The adjective used by the authors (“high” doses) indicates that the authors are biased about the side effect since the clinical dosage is according to the body weight to minimize adverse effects. I do not comprehend why adverse reaction is to be highlighted as a blessing in disguise.

The fourth discursive block “tuberculosis and Covid-19 – a necessary discourse” has a contextual relevance that has highlighted the barrier to seeking care. It highlights the impact of low preparedness of system on disease programs. More in depth analysis of this discursive block might have added more relevance to the study rather than the contribution to the currently available knowledge base by the three other blocks.

Reviewer #2: Thank you for the opportunity to read this interesting paper.

I find it well wrote on important research question.

Below my suggestions

1. Introduction:Update data on Tb burden from TB Report 2020. Line 100-107: add the role of social determinants of health in TB MDR onset and on therapy failure (see and cite Social determinants of therapy failure and multi drug resistance among people with tuberculosis: A review. Tuberculosis (Edinb). and Predictors of therapy failure in newly diagnosed pulmonary tuberculosis cases in Beira, Mozambique. BMC Res Notes. 2018 Feb 5;11(1):99.)

2.Methods and Results: are clear

3. Discussion: add the role of covid pandemic and disruption of Tb services resulting in TB diagnosis delay and worste clinical presentation and outcome

6. PLOS authors have the option to publish the peer review history of their article (what does this mean?). If published, this will include your full peer review and any attached files.

Reviewer #1: No

Reviewer #2: No

---

## [Author Response · Author response to Decision Letter 0]

19 Mar 2021

Dear Editor,

Thank you for the review of the manuscript “PONE-D-21-05519 - Patients’ perceptions regarding multidrug-resistant tuberculosis and barriers to seeking care in a priority city in Brazil during COVID-19 pandemic: A qualitative study”. 

We would like to thank the reviewers for their helpful and comprehensive comments, which clearly contributed to improving our study. 

The interview guided was uploaded as a Supplementary file as requested in this version of the manuscript. 

In this version submitted to the journal, we updated the question of Data Availability Statement. The data were collected upon request and approval by a Research Ethics Committee. The data has ethical restrictions on sharing a data set. 

In relation to Fig 1, it was constructed using data available from a Brazilian public repository, the Brazilian Institute of Geography and Statistics (IBGE) (https://www.ibge.gov.br/estatisticas/downloads-estatisticas.html), open access, without restrictions and without copyright. The data used for making the map (Fig 1) refer to files in the Shapefile extension, which were manipulated through the ArcGis® 10.6 software for making the maps, with the authors being entirely responsible for drawing the figure.

As requested, we present our responses to each point raised by reviewers. The changes made to the manuscript were highlighted in yellow color as requested. 

Best regards,

Santos et al. 

Reviewers' comments:

Reviewer #1

The aim of the study was to analyse the discourses of participants who were diagnosed with MDR-TB, the perception of why they acquired this health condition and barriers to seeking care during the Covid-19 pandemic. I tried to see whether the objectives have been met. 

Of the four discursive blocks formed, the first; “why did the disease develop? Social and economic factors” is expected to bring out the perceptions of the participants on the causal factors of MDRTB. Though the discursive analysis did bring out the fact that the participants have a low socioeconomic status, the verbatim did not suggest that the participants have perception that their low socioeconomic condition has causal relationship with their disease condition. The discursive analysis did not yield significant additional information than what could have been inferred from the quantitative data on the sociodemographic profile as four among the seven are unemployed.

Authors: Thank you for your comment. The title of the first discursive block was changed to “Impact of the social determinants in the development of multidrug-resistant tuberculosis”. We added verbatim after a more in depth analysis with the objective to bring out the socioeconomic perceptions of the participants on the causal factors of MDR-TB.

Page 16, line 343-344: “Impact of the social determinants in the development of multidrug-resistant tuberculosis” 

Page 17, line 349-350: “My son got sick first, but my house is not big, and we all sleep in the same room, I believe that’s why I got sick too” (P. 01)

Page 17, line 351 – 354: “Ah, during my first treatment, I lost my job, and I was losing a lot of weight. I work as a cleaner... As I was very weak, I was without work and I was unable to buy food for home. I was having help from my neighbors. I think that's why the first treatment didn't work...” (P. 04)

Page 17, line 355 – 359: “When I started feeling the first symptoms, I didn't want to believe that I was starting to get sick! I am the only person in my house who works ... when I was diagnosed with tuberculosis I spent the first few months without working ... the drugs made me very bad ... the doctor said that I needed to eat well to start gaining weight, but how could I eat well if I didn't have the enough money to buy food for my house?” (P. 05)

Page 17, line 360 – 362: “After six months he (son) had another one, then he took the exam and was diagnosed as having multidrug resistance. It was shocking for us who live... all together, so it was really hard!” (P. 07)

The second discursive block “barriers to seeking care and difficulties assessing health services” has met the objective.

Authors: Thank you for your comment. 

The third discursive block “a blessing in disguise: side effects of multi drug resistant tuberculosis treatment” has many weaknesses. The adjective used by the authors (“high” doses) indicates that the authors are biased about the side effect since the clinical dosage is according to the body weight to minimize adverse effects. I do not comprehend why adverse reaction is to be highlighted as a blessing in disguise.

Authors: Thank you for your comment. The title of the third discursive block has been changed to "Perceptions of the side effects and their impact on multidrug-resistant tuberculosis treatment". The paragraph was rephrased, the expression high doses was removed. We performed a new discursive analysis and added verbatim to make it more strengths.

Page 19, line 402 – 403: “Perceptions of the side effects and their impact on multidrug-resistant tuberculosis treatment”

Page 19, line 404 – 407: “The drugs used in the treatment of MDR-TB and the long period of exposure influence the appearance of side effects that affect the dynamics of adherence to treatment. It was observed that the research participants had side effects from the first few days of treatment.”

Page 19, line 412: “A little bit of myalgia. Arthralgia in the first treatment.” (P. 06)

Page 19, line 413 – 414: "Ah, when the treatment was here... I didn't gain weight and I was in a lot of pain." (P. 03)

Page 20, line 419 – 421: “Look, I have some numbness in my legs, you know, from the medicine. Even tomorrow, I have another return… It looks like they are going to decrease the medication already. They will decrease the dose, right...” (P. 01)

Page 20, line 431 – 432: “Then I talked to the doctor, then he said… so, we will then change your medication.” (P. 02)”

Page 20, line 436 – 440: “Ah I thought so, oh, what for? I don't have ... that lung is already lost. Why do I have to sacrifice myself going there, take these medications and it's not working? Huh. But then, I think like that, no ... if I'm going, if you're asking me to go, something is better, right. But then I took it out of my mind, then I already have something else to think about, right, that I have to continue to see what happens, right.” (P. 06)

Page 20, line 441 – 443: " In this sense, the nurse is very friendly, a counselor, she advised us not to give up, not to be discouraged, you know, to go ahead that everything would be all right.." (P. 04)

The fourth discursive block “tuberculosis and Covid-19 – a necessary discourse” has a contextual relevance that has highlighted the barrier to seeking care. It highlights the impact of low preparedness of system on disease programs. More in depth analysis of this discursive block might have added more relevance to the study rather than the contribution to the currently available knowledge base by the three other blocks.

Authors: Thank you for your comment. We added more verbatim after a more in depth analysis to make the fourth discursive block more relevance and giving an overall about the impact of the COVID-19 in the MDR-TB context. 

Page 21, line 458 – 459: “I was more afraid of COVID... because it is such a fast disease, you know.” (P. 04)

Page 21, line 460 – 462: “Look, like this, I try not to leave the house, I'm afraid to go out to get the medicine. I call the hospital when I have to get the medication, you know, they receive the prescription and send it to me, so I can stay isolated...” (P. 02)

Page 22, line 488 – 490: “Like this, I stay inside my house a lot ... But every now and then I feel like going to my mom's house… but I know that it is complicated to visit her! In addition to COVID, I am afraid that she will also become ill due to tuberculosis” (P. 05)

Reviewer #2.

1. Introduction: Update data on Tb burden from TB Report 2020. 

Authors: Thank you for your comment. All the data on TB burden was updated from TB Report 2020. 

Page 3-4, line 67 – 76: “According to the World Health Organization (WHO), there were an estimated 465,000 cases of MDR-TB in 2019, equivalent to 11% of the total number of cases of TB worldwide. In the same year, an estimated 3.3% of new TB cases and 18% of previously treated cases had MDR-TB [1]. According to the latest data reported in MDR-TB annual cohorts, of those affected by the disease who started treatment for MDR-TB, only 57% completed the treatment successfully, 16% were lost to follow-up, 7% had treatment that failed, and 15% died, while for 5% there was no outcome information [1]. When these findings are compared with the outcomes in new or recurrent cases of sensitive TB (85% cure), they are very low for those with MDR-TB, highlighting one of the main global challenges of the disease [1].”

Page 4, line 77 – 82: “In 2019, the total number of people diagnosed with TB who underwent specific tests for MDR-TB was higher than in previous years. Worldwide, 81% of previously treated patients and 59% of new cases were tested for MDR-TB, which was an increase of 10% and 61%, respectively, compared to 2018. This increase is mainly due to the implementation of the GeneXpert MTB/RIF assay and improved laboratory diagnostics [1].”

Line 100-107: add the role of social determinants of health in TB MDR onset and on therapy failure (see and cite Social determinants of therapy failure and multi drug resistance among people with tuberculosis: A review. Tuberculosis (Edinb). and Predictors of therapy failure in newly diagnosed pulmonary tuberculosis cases in Beira, Mozambique. BMC Res Notes. 2018 Feb 5;11(1):99.)

Authors: Thank you for your comment. We added new paragraphs in the manuscript. The role of Social Determinants of Health in MDR-TB onset and on therapy failure were added referencing the two studies. 

Page 5, line 108 – 113: “In addition, the poor educational degree has an important role in the onset and in the MDR-TB treatment loss to follow-up. The low educational level, as well as low-income, is associated with poor infrastructures, limiting the access to health services. Some behavioral issues, such as alcohol abuse, are also predictors that determine the treatment successfully or treatment failure outcomes. The low-income groups tend to have higher levels of related to behavioral issues harm than high-income groups [12, 13].” 

Page 5, line 114 – 117: “Social protection interventions are relevant strategies that support to reduce poverty, and consequently, provide better treatment outcomes, with additional effects in reducing MDR-TB cases [12, 13]. These determinants have a great influence on patients seeking care as well their adherence to treatments and prescription regimes [14].”

2. Methods and Results: are clear 

Authors: Thank you for your comment. 

3. Discussion: add the role of covid pandemic and disruption of Tb services resulting in TB diagnosis delay and worste clinical presentation and outcome.

Authors: Thank you for your comment. New paragraphs were inserted in the discussion section. The role of COVID-19 in the TB health services was added and the importance to maintain and strengthen TB services as an essential component of overall progress towards universal health coverage and resilient health systems, and to ensure synergies in the responses to both TB and COVID-19. 

Page: 26, line 583 – 586: “The pressure on health services resulting from the COVID-19 pandemic, combined with impacts on care-seeking behavior, could slow or reverse progress towards TB treatment and prevention targets set at the United Nation high-level meeting on TB, especially in high TB burden countries [1].”

Page 27, line 594 – 597: “In the first two months of 2020 was identified reductions in TB-related hospital discharges, newly diagnosed cases of active TB, total active TB outpatient visits, and new latent tuberculosis infection (LTBI) cases and LTBI outpatient visits. These results may be explained by a general decrease in the use of health services [59].”

Page 27, line 598 – 604: “Resources for TB service provision were reassigned to other medical services. Outpatient visit numbers may have decreased because of patients’ fear of exposure to COVID-19 [60]. Access to health services may have decreased because of interruptions in or difficulty accessing public transportation, although health-related travel was permitted in most countries [61]. There is already evidence from several high TB burden countries of large reductions in the monthly number of people with TB being detected and officially reported in 2020 [1].”

Page 28, line 624 – 626: “It is crucial to maintain and strengthen TB services as an essential component of overall progress towards universal health coverage and resilient health systems, and to ensure synergies in the responses to both TB and COVID-19 [1].”

---

## [Decision Letter · Decision Letter 1]

26 Mar 2021

Patients’ perceptions regarding multidrug-resistant tuberculosis and barriers to seeking care in a priority city in Brazil during COVID-19 pandemic: A qualitative study

PONE-D-21-05519R1

Dear Dr. Felipe,

We’re pleased to inform you that your manuscript has been judged scientifically suitable for publication and will be formally accepted for publication once it meets all outstanding technical requirements.

Kind regards,

Francesco Di Gennaro

Academic Editor

PLOS ONE

Additional Editor Comments (optional):

congratulations

Reviewers' comments:

Reviewer's Responses to Questions

**Comments to the Author**

1. If the authors have adequately addressed your comments raised in a previous round of review and you feel that this manuscript is now acceptable for publication, you may indicate that here to bypass the “Comments to the Author” section, enter your conflict of interest statement in the “Confidential to Editor” section, and submit your "Accept" recommendation.

Reviewer #1: All comments have been addressed

Reviewer #2: All comments have been addressed

2. Is the manuscript technically sound, and do the data support the conclusions?

Reviewer #1: Yes

Reviewer #2: Yes

3. Has the statistical analysis been performed appropriately and rigorously? 

Reviewer #1: Yes

Reviewer #2: Yes

4. Have the authors made all data underlying the findings in their manuscript fully available?

Reviewer #1: Yes

Reviewer #2: Yes

5. Is the manuscript presented in an intelligible fashion and written in standard English?

Reviewer #1: Yes

Reviewer #2: Yes

6. Review Comments to the Author

Reviewer #1: Thanks for revising the manuscript considering the suggestions. I do not have additional comments on the revision.

Reviewer #2: authors improved their paper that now can be accept. The research question is important and also the setting of study is relevant. congratulations

7. PLOS authors have the option to publish the peer review history of their article (what does this mean?). If published, this will include your full peer review and any attached files.

Reviewer #1: **Yes: **Shibu Balakrishnan

Reviewer #2: No

---

## [Editor Report · Acceptance letter]

1 Apr 2021

PONE-D-21-05519R1 

Patients’ perceptions regarding multidrug-resistant tuberculosis and barriers to seeking care in a priority city in Brazil during COVID-19 pandemic: A qualitative study 

Dear Dr. Santos:

I'm pleased to inform you that your manuscript has been deemed suitable for publication in PLOS ONE. Congratulations! Your manuscript is now with our production department. 

Kind regards, 

on behalf of

Dr. Francesco Di Gennaro 

Academic Editor

PLOS ONE